# Predicting Risk of Bullying Victimization among Primary and Secondary School Students: Based on a Machine Learning Model

**DOI:** 10.3390/bs14010073

**Published:** 2024-01-20

**Authors:** Tian Qiu, Sizhe Wang, Di Hu, Ningning Feng, Lijuan Cui

**Affiliations:** 1Shanghai Key Laboratory of Mental Health and Psychological Crisis Intervention, Institute of Brain and Education Innovation, School of Psychology and Cognitive Science, East China Normal University, Shanghai 200062, China; qiutianecnu@163.com; 2School of Statistics, East China Normal University, Shanghai 200062, China; 52264404023@stu.ecnu.edu.cn; 3Sliver School of Social Work, New York University, New York, NY 10012, USA; dh2843@nyu.edu; 4Shanghai Centre for Brain Science and Brain-Inspired Technology, Shanghai 200062, China

**Keywords:** bullying victimization, school bully, ecological factors, longitudinal study, machine learning, GBDT

## Abstract

School bullying among primary and secondary school students has received increasing attention, and identifying relevant factors is a crucial way to reduce the risk of bullying victimization. Machine learning methods can help researchers predict and identify individual risk behaviors. Through a machine learning approach (i.e., the gradient boosting decision tree model, GBDT), the present longitudinal study aims to systematically examine individual, family, and school environment factors that can predict the risk of bullying victimization among primary and secondary school students a year later. A total of 2767 participants (2065 secondary school students, 702 primary school students, 55.20% female students, mean age at T1 was 12.22) completed measures of 24 predictors at the first wave, including individual factors (e.g., self-control, gender, grade), family factors (family cohesion, parental control, parenting style), peer factor (peer relationship), and school factors (teacher–student relationship, learning capacity). A year later (i.e., T2), they completed the Olweus Bullying Questionnaire. The GBDT model predicted whether primary and secondary school students would be exposed to school bullying after one year by training a series of base learners and outputting the importance ranking of predictors. The GBDT model performed well. The GBDT model yielded the top 6 predictors: teacher–student relationship, peer relationship, family cohesion, negative affect, anxiety, and denying parenting style. The protective factors (i.e., teacher–student relationship, peer relationship, and family cohesion) and risk factors (i.e., negative affect, anxiety, and denying parenting style) associated with the risk of bullying victimization a year later among primary and secondary school students are identified by using a machine learning approach. The GBDT model can be used as a tool to predict the future risk of bullying victimization for children and adolescents and to help improve the effectiveness of school bullying interventions.

## 1. Introduction

When a student is exposed to the negative behavior from one or more other students repeatedly and over a long period, this is identified as being bullied or victimized [1]. School bullying is prevalent worldwide. Data from school bullying surveys in 144 countries and regions indicated that one-third of young people worldwide have experienced bullying in the past two months [2]. In China, a report showed that 33% of students in rural areas and 25.8% in urban areas have experienced bullying victimization [3]. Bullying victimization can lead to adverse consequences, including difficulties in adaptation and psychosocial functioning [4]. Furthermore, bullying victimization by peers in childhood has generally worse long-term detrimental effects on mental health in young adults, including anxiety, depression and self-harm [5]. Therefore, it is necessary to identify predictors of bullying victimization to protect the positive development of children.

Previous researchers have explored the factors related to bullying victimization at school, revealing a complex array of influencing elements. Bullying victimization is associated with both the individual factors and the ecosystem in which students occur. Individual factors related to student victimization include students’ psychological characteristics and emotional issues [6]. Additionally, microsystem factors such as family, school, and peers also significantly predict an individual’s risk of bullying victimization [7,8]. However, previous research is mainly based on a single factor or interplay between several factors, lacking comprehensive and systematic empirical evidence.

Machine learning (ML) methods can provide solutions superior to traditional statistical methods for problems affected by multiple factors, especially for extensive sample data. Some learning algorithms can perform automatic feature selection. For example, the GBDT algorithm can screen out the most important features for large, high-dimensional datasets. This allows for more predictors and considerably reduces the interference of human factors. Moreover, machine learning models emphasize the predictive accuracy of output models. In some studies, ML models are more accurate than traditional regression models [9,10,11]. Finally, some categories of learning algorithms are inherently interpretable. For example, decision tree algorithms generate simple conditional rules (e.g., “if gender is female and age is over 42, then the predicted value is 1”), which are more intuitively understandable than continuous value prediction equations generated by regression methods [12].

## 2. Literature Review

According to the ecosystem theory [13], an individual behavior is influenced by both their personal characteristics and the ecosystem they are in. Individuals’ immediate environment for contacts and interactions is the microsystem, which has the most direct and frequent impact on them among all ecosystems. Based on previous research, there are four main ecological factors related to bullying victimization at school: individual, family, peers, and school. 

### 2.1. Individual Factors and Bullying Victimization

Previous studies have found that students who have been bullied in school share common psychological or behavioral characteristics: These students typically demonstrate higher levels of neuroticism, irritability, and impulsivity compared to their peers who are not bullied [14]. Moreover, they tend to exhibit more pessimistic emotions [15], display lower levels of self-compassion and hope [16,17], and are more inclined to use avoidance strategies in dealing with challenges [18]. In addition, studies have found that mental health problems are one of the significant factors associated with bullying victimization, and adolescents with depression are more likely to develop difficulties in peer relationships and are more vulnerable to being bullied at school [19,20]. Moreover, insecurity, anger, low self-control, loneliness, and high social anxiety are also closely related to bullying victimization [21,22,23,24,25]. Di Blasio’s [26] study showed that heightened psychological vulnerability in individuals could lead to a greater likelihood of being targeted by bullies.

Within the context of demographic factors, left-behind children, whose parents work outside the home more than six months, are often considered as vulnerable people. Previous research has found that left-behind children are more likely to suffer from peer bullying than non-left-behind children [27], and boys are more likely to be involved in physical or verbal bullying than girls [28]. Additionally, the student’s grade level is also an important factor, and the incidence of peer victimization tends to decrease as students advance in grade level [29].

### 2.2. Family Factors and Bullying Victimization

Parenting styles and the quality of parent–child relationships in the family are associated with bullying victimization at school. Patton et al. [30] indicated that children exposed to harsh or abusive parenting methods are more susceptible to involvement in school bullying. Poor parent–child relationships, as well as inadequate communication, can increase the likelihood of children becoming victims of bullying [31]. On the other hand, a positive family atmosphere and warmth from family members can reduce children’s risk of bullying victimization [32]. Furthermore, parental control is significantly associated with a higher chance of experiencing physical attacks [33].

### 2.3. Peer Factors and Bullying Victimization

Primary and secondary school students spend a significant portion of their day with peers. Peer factors play an important role in bullying and bullying victimization. Research has found that good peer relationships could reduce bullying incidence [30].

On the other hand, being a victim of bullying has been found to be associated with a lack of support from peers [34]. In line with this, a study also showed that early adolescent victims tend to experience heightened conflicts with their closest companions and encounter greater challenges when it comes to managing confrontations with friends [35].

### 2.4. School Factors and Bullying Victimization

Teachers are in the best position to understand the school situation of their students and can thus play a crucial role in addressing bullying in schools. When students encounter any challenges, teachers can provide them with practical help. Previous studies have found that adolescents are less likely to be bullied when they perceive a more harmonious teacher–student relationship [36].

In addition, children with learning difficulties are at increased risk of bullying victimization [37]. Students with poor school adaptability and slower completion of learning tasks are more likely to be bullied [38].

Based on previous research, we hypothesized that individual, family, peer, and school factors are significant predictors of bullying victimization among primary and secondary students.

### 2.5. Machine Learning in Research on Bullying

With the development of machine learning technology, some empirical studies in psychology have attempted to apply machine learning algorithms to predict problem behaviors. It has been observed that robust tree-based classification algorithms can effectively predict such behaviors [9,39,40], which can more accurately divide individuals into non-risk and risk groups, especially when the risk assessment tool for problem behaviors is predetermined.

School bullying is a type of problem behavior that can be classified and predicted through the use of machine learning. Previous studies have also explored machine learning methods to predict the risk of bullying victimization. For example, Hani et al. [41] used a Neural Network to recognize cyberbullying actions and achieved accuracy of 92.8%. Dalvi et al. [42] used support vector machines and naive Bayes algorithms to detect and prevent bullying on social media platforms like Twitter. However, most of these studies have mainly focused on identifying and predicting cyberbullying phenomena rather than actual school bullying behaviors.

In addition, integrated machine learning algorithms have shown superior predictive accuracy in classification tasks. One of the most representative algorithms is the gradient boosting decision tree (GBDT). It is a tree-based algorithm that can combine multiple weak classifiers to form a strong one, which can achieve more accurate classification results. GBDT can also assess the importance of input features based on the purity gain from each tree split. Therefore, the aim of this study is to utilize the machine learning algorithm of GBDT to predict the risk of bullying victimization one year later among primary and secondary school students. Additionally, it seeks to identify key risk and protective factors influencing bullying victimization among primary and secondary school students

## 3. Methods

### 3.1. Procedure

We recruited Chinese students from 9 primary and 6 secondary schools in our study at Time1 (T1, November 2020). We selected schools scattered across provinces in China that were less affected by COVID-19 pandemic control policies. These included schools in five provinces: Hebei, Henan, Jiangxi, Hubei, and Sichuan. Cluster sampling was conducted by class. Participants were asked to complete the scales measuring predictors of bullying victimization at T1, and then completed the scale of outcome variable a year later (T2, November 2021).

This study was approved by the University Committee on Human Research Protection (HR2-1018-2020). Before data collection, participants and their parents signed a paper consent form. All participants were informed that their participation was entirely voluntary and that they could withdraw at any time.

### 3.2. Participants

At T1, the sample consisted of 3015 students (47.03% boys, 52.97% girls; 39.49% primary school students, 60.51% secondary school students; Mage = 12.66, SDage = 2.17). At T2, 2767 participants of those who participated in the survey at T1 completed the questionnaire of the outcome variable (91.77% of T1, 44.80% boys, 55.20% girls; 25.37% primary school students, 74.63% secondary school students; Mage = 13.22, Sdage = 1.63).

### 3.3. Measures

Based on the ecological theory framework and previous research results, this study selected 15 mature scales to measure predictors (see Table 1 for the specific scales). The scales include three types: the Chinese translation of the English original version, the Chinese revised version of the English original version, and the Chinese original version. Predictors include four demographic variables: gender, grade, place of origin, and whether the participant was left behind.

For the outcome variable, the victim subscale of the Olweus Bully/Victim Questionnaire [43] was used to measure whether students were bullied. There are seven questions, including “Some classmates have called me ugly names, or made fun of and satirized me in the last year”. The questionnaire is scored using Likert’s five-point method: 1 = “never”, 2 = “once or twice”, 3 = “two or three times a month”, 4 = “about once a week”, and 5 = “several times a week”. The Cronbach α coefficient of the scale in our study was 0.855.

**Table 1 behavsci-14-00073-t001:** Summary of measures for predictors.

Type	Variables	Measures	Reliability (Cronbach’s Alpha)	Source of English Version	Source of Chinese Version
Individual Factors	Positive and Negative Affect	Positive and Negative Affect Scale	0.827	Bradburn, 1969 [44]	Chen and Zhang, 2004 [45]
	Feeling of Security	Security Questionnaire	0.910		Cong and An, 2004 [46]
	Anger	The Aggression Questionnaire	0.855	Buss and Perry, 1992 [47]	Li et al., 2011 [48]
	Hostility		0.830		
	Self-Control	Tangney Self-Control Scale	0.715	Unger et al., 2016 [49]	Luo et al., 2021 [50]
	Loneliness	Children’s Feeling of Loneliness and Social Dissatisfaction Scale	0.888	Asher et al., 1984 [51]	Wang et al., 1999 [52]
	Social Anxiety	Social Anxiety Scales	0.950	La Greca and Lopea, 1998 [53]	Wang et al., 1999 [52]
	Coping Style	Coping Style Scale for Middle School Students	0.856		Huang et al., 2000 [54]
	Self-Compassion	Self-Compassion Scale	0.707	Neff, 2003 [55]	Chen et al., 2011 [56]
	Hope	Children’ s Hope Scale	0.800	Snyder, 1997 [57]	Zhao et al., 2011 [58]
	Depression	The Depression Anxiety Stress Scale	0.870	Antony et al., 1998 [59]	Gong et al., 2010 [60]
	Anxiety		0.808		
	Stress		0.824		
Family Factors	Family Cohesion	Family Adaptability and Cohesion Scale	0.925	Olson et al., 1982 [61]	Wang, 2015 [62]
	Cold Parenting Style	Parental Bonding Instrument	0.913	Parker et al., 2011 [63]	Jiang et al., 2009 [64]
	Denying Parenting Style		0.931		
	Parental Control	Perceived Parental Control Scale	0.854	Shek, 2000 [65]	Liu, 2021 [66]
Peer Factors	Peer Relationship	Social Adaptation Questionnaire for Children	0.951		Guo et al., 2018 [67]
School Factors	Teacher–Student Relationship	Middle School Student’s Social Adaptation Scale	0.807		Yang, 2007 [68]
	Learning Capacity		0.764		

### 3.4. Analytic Method

#### 3.4.1. Data Processing

We adopted the GBDT algorithm to predict whether participants would be bullied a year later. We used the K-Nearest Neighbor algorithm to fill in missing data by referencing similar samples.

The dependent outcome variable (i.e., bullying victimization) turned out to be a categorical variable; 0 represents “the individual had not been bullied in the past year” and 1 represents “the individual had been bullied in the past year”. Among the 7 questions of victim subscale, the participants who scored ranging from 2 to 5 on at least one question were coded as the group “1” (bullying victimization); the participants who scored 1 on all questions were coded as the group “0” (not bullying victimization). Finally, 1786 participants (64.55%) were grouped as bullying victimization (group 1) and 981 samples participants (35.45%) of were grouped as non-bullied (group 0). The proportion of the sample two groups was not balanced, which may affect the accuracy of machine learning model prediction. Therefore, the smote oversampling technique was used to expand a small number of samples to generate balanced data samples.

The data were divided in the ratio of the training set: test set = 8:2. The training dataset was used in the model-building process and the test dataset was used in the model testing process.

In addition, in order to compare the performance of GBDT the algorithm and traditional logistic regression method, we used SPSS 21.0 to perform linear logistic regression analysis. In the linear logistic regression analysis, the outcome variable was the bullying victimization (1 = bullying victimization, 0 = non-bullied) and the predictors were the same as in the ML analysis.

#### 3.4.2. Feature Selection and Prediction Algorithm

The mean score of each independent variable’s scale (or subscale) was used as an input feature, and 29 features were modeled.

Our study used the GBDT algorithm from the machine learning modeling algorithm package in Python for modeling. GBDT uses the boosting integrated learning method. Specifically, the boosting idea is that training a base learner in the initial training set, and then adjusting the training sample weights based on the performance of the previous learner. The training samples that the previous base learner performed incorrectly are subsequently given increased weight, and then the next base learner is trained based on the adjusted sample distribution. GBDT’s base learner is a decision tree. The decision tree is a supervised learning algorithm based on a tree structure often used to deal with classification tasks. As an integrated algorithm, GBDT has a stronger ability to predict different classifications than a decision tree.

The GBDT model is trained using the training dataset, and the outcome variable of the test dataset is predicted using the model. By comparing the actual classification results of the test set with the model’s prediction results on the test set, we can obtain the predicted two-dimensional confusion matrix for this binary classification study. The constituent elements of the confusion matrix include (1) TP (i.e., the number of samples with predicted value of 1 and true value of 1), (2) TN (i.e., the number of samples with the predicted value of 0 and true value 0), (3) FP (i.e., the number of samples with predicted value 1 but the true value 0), and (4) FN (i.e., the number of samples with predicted value 0 but true value 1).

Five measures of model prediction can be computed from the confusion matrix: accuracy = (TP + TN)/(TP + TN + FP + FN), precision = TP/(TP + FP), recall = TP/(TP + FN), specificity = TN/(TN + FP), and F1 = 2/(1/precision + 1/sensitivity). The F1 indicator is the summed average of precision and recall, which effectively indicates the classification accuracy of the model. The ROC curve can also be plotted based on the confusion matrix of the model classification prediction. The ROC curve effectively reflects the generalization ability of the learner and can help researchers adjust classification thresholds to meet the specific needs of the model. The vertical axis of the ROC curve is the “true case rate” and the horizontal axis is the “false positive case rate”. The AUC value is the area under the ROC curve, and the AUC generally ranges from 0.5 to 1. An AUC value close to 1 indicates a better classifier, whereas an AUC value of 0.5 indicates a poor classifier and is not valuable for application. Logistic regression can also obtain two-dimensional confusion matrix and six indicators (i.e., accuracy, precision, recall, specificity, F1 and AUC). Following the recommendation in existing literature, these indicators were regarded as criteria of the predictive performance of the GBDT algorithm and traditional logistic regression in our study [69].

If the model performance is not satisfactory, the model performance can be improved by adjusting the hyperparameters of the model. The main primary hyperparameters of the GBDT algorithm include (1) learning_rate (it controls the contribution of each tree), (2) n_estimators (the number of boosting stages to perform), (3) max_depth (the maximum depth of the individual regression estimators), (4) min_samples_leaf (the minimum number of samples required to be at a leaf node), (5) min_samples_split (the minimum number of samples required to split an internal node), (6) max_features (the number of features to consider when looking for the best split), and (7) subsample (the fraction of samples to be used for fitting the individual base learners).

## 4. Results

### 4.1. Sample Description

Table 2 presents the individual, family, peer, and school factors for the “bullying victimization group” and the “not bullying victimization group”. More than 60% of the surveyed students had been bullied, indicating that bullying victimization was common among primary and secondary school students. Moreover, there were significant differences between the individual, family, peer and school factors for the students who had been bullied and those who had not been bullied.

### 4.2. Classification Results

Table 3 presents the modeling prediction of bullied risk using the GBDT method and logistic regression analysis. We chose the model parameters as learning_rate = 0.1, n_estimators = 500, max_depth = 11, min_samples_leaf = 5, min_samples_split = 10, max_features = 19, random_state = 123, and subsample = 0.8. The accuracy of the model using the GBDT method reached 79.58%. Figure 1 shows the ROC curve and the AUC value of this model was 0.86. For the logistic regression model, the accuracy and AUC were both 0.67. Therefore, the GBDT algorithm performed better in predicting the risk of bullying victimization than the traditional logistic regression in our study.

### 4.3. Feature Importance

Figure 2 shows the importance of each feature in the GBDT model for determining the outcome (i.e., the decision classification of whether or not to be bullied a year later). The feature importance score indicates the relative importance of the features used in the GBDT model in predicting the target variable. The importance of each feature is calculated as the average of the feature importance of each decision tree. The feature importance of each tree is evaluated based on the amount of purity improvement of the nodes before and after splitting, calculated as Mean Square Error (MSE) or Mean Absolute Error (MAE). When this feature is used as the partition node of the decision tree, the higher the feature importance score, the higher the accuracy of the classification result for the dependent variable. Feature importance can help us understand which features play a crucial role in the model.

In our study, 22 individual features varied widely in importance. Negative affect and anxiety were the crucial predictors (score = 0.043), ranked 4th and 5th, respectively. Depression, self-compassion, anger, social anxiety, and hostility also occupied crucial positions (score = 0.038/0.037). Except for self-compassion, the other positive features ranked lower in importance: security scored 0.03, self-control scored 0.03, and hope scored 0.03. Demographic variables scored low in feature importance: Gender scored 0.029, grade scored 0.02, place of birth scored 0.013, and left-behind children or not scored 0.011.

As for four family features, the importance of family cohesion (score = 0.045) ranked third among all predictors. Denying parenting style was also found to be an important predictor (score = 0.043). Parental control (score = 0.033) and cold parenting style (score = 0.03) were found to be less important than family cohesion and denying parenting style.

For school and peer factors, teacher–student relationship (score = 0.059) and peer relationship (score = 0.045) were crucial predictors. In particular, teacher–student relationship scored significantly higher than other predictors in the GBDT model. Learning capacity was less important (score = 0.037).

Although the exact quantitative criterion is still lacking, ML studies usually focus on the top N predictors of feature importance based on research goals [69,70]. This study focused on the top 6 predictors with the highest feature importance. Among them, teacher–student relationships, family cohesion and peer relationships were protective factors. Negative affect, anxiety, and denying parenting style were identified as risk factors.

## 5. Discussion

Using a machine learning approach, this longitudinal study aimed to identify the risk and protective factors for school bullying among primary and secondary school students. Our study examined 29 features including individual, family, peer, and school factors, to predict the risk of bullying victimization using the gradient-boosting decision tree model (GBDT) one year later. In this study, the accuracy of predicting the risk of bullying victimization over a year for primary and secondary school students using the GBDT model reached 79.58%, higher than the result of logistic regression (67.08%). This showed that the GBDT method was superior to traditional logistic regression in our study. This study’s findings contribute to understanding the risk and protective factors associated with school bullying. Our study also provides a fresh tool to predict the future risk of bullying victimization.

Our study demonstrates another benefit of the GBDT algorithm. The GBDT model identified six strong predictors: teacher–student relationship, peer relationship, family cohesion, negative affect, anxiety, and denying parenting style. The GBDT algorithm can handle high-dimensional data with a large number of variables and can identify the most important features that are relevant to the response variable. The multicollinearity problem has a milder effect on the GBDT algorithm, so the GBDT algorithm is more robust [71]. Traditional logistic regression is more susceptible to the effects of multicollinearity, which reduces the stability and interpretability of the model, and may not be suitable for high-dimensional data with a large number of variables. Overall, the GBDT algorithm is a powerful data analysis tool in selecting essential variables.

For individual features, our study found that negative psychological characteristics were more powerful than positive psychological characteristics in predicting whether children would be bullied later. Higher levels of internalizing problems have been found to predict victimization in childhood and adolescence in other research [72]. According to the attention bias theory, negative emotions and behaviors are more likely to attract others’ attention [73]. Thus, individuals with negative psychological characteristics are more likely to have experienced peer victimization because they are more likely to be noticed and chosen as targets by the attackers. Moreover, it was a positive sign that hard-to-change demographic factors did not have a significant impact on whether children would be bullied.

The findings indicated that family and school factors played significant roles in predicting the risk of bullying victimization for students. If we look at the positive and negative aspects of predictive factors, this point could also be corroborated.

In our study, the top three positive factors were teacher–student relationship, peer relationship, and family cohesion. They were all factors in the relationship between students and their significant others. Focusing on each predictor concretely, a solid and positive teacher–student relationship fosters a sense of belonging to the school, which helps prevent bullying [74]. When students have support from their teachers, they feel recognized and accepted, and they can develop a good psychological connection to the school. If students are involved in school bullying, they are more likely to report it to their teachers, thus blocking the influence of risk factors on students and effectively reducing their probability of bullying victimization. On the other hand, poor teacher–student relationships can leave students feeling rejected. With an unmet sense of belonging, it is also difficult for students to seek help from teachers when they become the target of school bullying, thus suffering more school bullying.

Similarly, peer relationship was a significant predictor of peer victimization for elementary and middle school students. Previous studies also found that children with positive relationships with their peers were less likely to be bullied [7,75]. This suggested that peer care could help adolescents cope with school bullying incidents. Good connections with peers can provide adolescents with sufficient emotional support to counteract the risk of bullying victimization. Nevertheless, social isolation is a factor that poses a heightened risk for victimization in adolescence. Those adolescents who are unable to fit into one or multiple peer groups are more likely to become targets of victimization by their peers [76].

The relationship between children and parents, particularly parental involvement and support, was a key protective factor against bullying. According to previous research [77], parents provide adolescents with social skills and appropriate interpersonal relationships to promote the development of the whole self. Parents involved in their children’s lives can provide adequate emotional support, model positive behaviors, and strengthen their children’s social and emotional strategies. Thus, they can help their children develop healthy relationships with others and prevent malignant incidents of bullying victimization.

The important predictive role of microsystem factors also provided positive signals for interventions for children exposed to bullying. School, peer, and family factors allow for more direct and effective interventions than stable individual characteristics. Interventions and support measures that target factors such as peer, school, and family environments can help individuals improve interpersonal relationships, enhance self-protection, and promote emotional regulation to avoid the risk of bullying victimization.

Our study found that negative affect, anxiety and denying parenting styles were top three risk factors in predicting whether primary and secondary students would be bullied. An individual’s emotions and affects are valuable sources of information that allow individuals to quickly assess their environment and adaptively guide their subsequent actions [78]. When students are in a negative or anxious mood, they often behave in ways that demonstrate low self-esteem, insecurity, and hostility. These behaviors may be perceived by bullies as signs of weakness. Moreover, during the developmental stage where many youths try to assert dominance by targeting weaker peers, displaying negative emotions and anxiety might inadvertently mark some as vulnerable, thereby increasing their risk of facing peer aggression [79]. In addition, children with emotional problems and symptoms of anxiety may be oversensitive to the intentions and behaviors of their peers. They may interpret thoughtless actions and words as aggressive and hostile, leading to misunderstandings and resentment toward other children. This situation may lead to conflicts and clashes between children and even cause bullying. Worse yet, children with mental health issues may lack the ability to defend themselves and cope with bullying. When they are bullied, they may feel helpless and scared and unable to respond and solve problems effectively. These children might lack effective coping mechanisms for bullying, exacerbating and perpetuating the situation.

Denying parenting style could also increase the risk of bullying victimization for primary and secondary students. Denying parenting style manifests as a lack of attention, indifference, scolding, and punishment, which may cause children to feel unappreciated and unsupported by their parents and affect their social skills and interpersonal interactions [80], resulting in a lack of close relationships and friend support at school and among their peers. Meanwhile, denying parenting style may increase the risk of adolescent maladjustment, which leads to their vulnerability to negative affect (e.g., mood dysregulation, depression, anxiety) and aggression [81], thus affecting the child’s attitudes and behavioral performance toward others and making them more vulnerable to peer rejection and bullying victimization.

## 6. Limitations and Suggestions for Future Studies

Firstly, our study demonstrates that the GBDT model is a compelling method of data analysis for investigating behavioral problems that are influenced by multiple and complex factors. However, the potential accuracy and ethical issues merit attention when using machine learning as a predictive tool for at-risk people. Using algorithms may be harmful, especially when machine learning applications make incorrect diagnoses or predictions. Previous limitations in health-related machine learning can guide in this regard [82]. Additionally, should participants be asked for their explicit consent before providing data to algorithms for predictive analytics? This question arises because the inherent opacity of machine learning can potentially conflict with the informed consent rights of participants. It is important to note that obtaining informed consent for the use of predictive analytics is currently not mandatory by law [83]. Furthermore, discussing machine learning algorithms with vulnerable groups may carry the risk of additional psychological pressure and may worsen their situation [84]. Future researchers should be cautious in using machine learning methods.

Secondly, although we included 29 predictive features in our analysis, other important factors need to be considered in the future. For instance, our study sample consisted of ordinary primary and secondary school students from China, and thereby, we did not collect information on nationality, race, and disability. However, these variables are crucial for studies in multicultural contexts, especially for minority populations [85,86]. Future studies should include more diverse sample populations. Furthermore, we did not ask the participants to report their previous bullying perpetration in this study. A meta-analysis has suggested that previous bullying perpetration among students is one of the most strongly correlated factors with subsequent bullying victimization [87]. Future studies should consider collecting details related to participants’ previous bullying perpetration. In addition, our study did not examine the interaction effects of the predictive features on primary and secondary school students’ risk of bullying victimization. However, according to person-context interaction theory [88], contextual factors (e.g., peer relationships) may interact with individual factors (e.g., anxiety) to influence adolescents’ behavior. Future studies could include more interactive variables to measure their effects.

Thirdly, our study did not differentiate the types of bullying victimization (e.g., verbal, relational, physical, cyber), which may be related to different predictive factors. Previous studies have found that physical bullying victimization is associated with family domains (i.e., parenting style). In contrast, relational and verbal bullying victimization is associated with multiple social domains (e.g., not being an only child, poor relationships with classmates) [89]. Future research can provide a more detailed analysis of different types of bullying victimization and explore whether different types have distinct predictive factors. This would help understand the nuances and specific predictive factors associated with each type of bullying victimization.

Fourthly, our study did not further differentiate the bullying victim group. Future research could subdivide the bullied group by the severity of bullying victimization (e.g., severely bullied group, mild bullied group) and explore whether there are differences in the predictive mechanisms of different types of bullied groups.

Finally, some predictive factors used for machine learning in our study may result from prior bullying victimization. For example, anxiety, stress, and depression, which are psychological health issues, have been considered as consequences of bullying victimization in other studies [90]. This may suggest that the predictive effect of these factors on bullying victimization reflects the association between previous experiences of victimization and subsequent victimization. Therefore, future research could utilize methods like cross-lagged analysis to explore further the longitudinal relationship between psychological health factors and bullying victimization.

## 7. Implications

Our study used a machine learning approach to systematically examine factors associated with the risk of bullying victimization one year later among primary and secondary school students. By utilizing the GBDT, our study showed the different roles of individual, family, school, and peer factors in predicting the likelihood of bullying victimization a year later. These findings contributed to the understanding of bullying victimization. Moreover, the application of the GBDT model as a predictive tool offered the potential for improving the accuracy of identifying children and adolescents at risk of bullying victimization.

Our study can assist in developing effective school bullying prevention and intervention strategies. Our study found that teacher–student relationships, peer relationships, and family cohesion played crucial roles in predicting the future risk of bullying victimization for primary and secondary school students, suggesting that efforts to improve the psychosocial environment at the systemic level are necessary. For example, teachers and students at school should be provided with relevant education and training to help them identify and prevent bullying and know how to respond. Parents need to maintain good communication with their children, understand their situation at school, and provide timely support to detect and address bullying; schools and students’ parents can work together to build a defense mechanism against bullying in schools. For example, schools can join parents in school anti-bullying actions and help parents understand the school’s anti-bullying policies and measures.

## 8. Conclusions

This study investigated the factors that may predict the risk of bullying victimization for primary and secondary school students. The findings revealed that teacher–student relationships, peer relationships, and family cohesion emerged as crucial protective factors. In contrast, negative affect, anxiety, and denying parenting style were identified as significant risk factors. These results highlight the importance of fostering positive relationships within the school environment and promoting supportive family dynamics to mitigate the risk of bullying. Furthermore, the GBDT model, as an effective predictive tool, holds promising prospects for identifying students at risk of future bullying victimization and for enhancing the efficacy of interventions to address school bullying issues.

## Figures and Tables

**Figure 1 behavsci-14-00073-f001:**
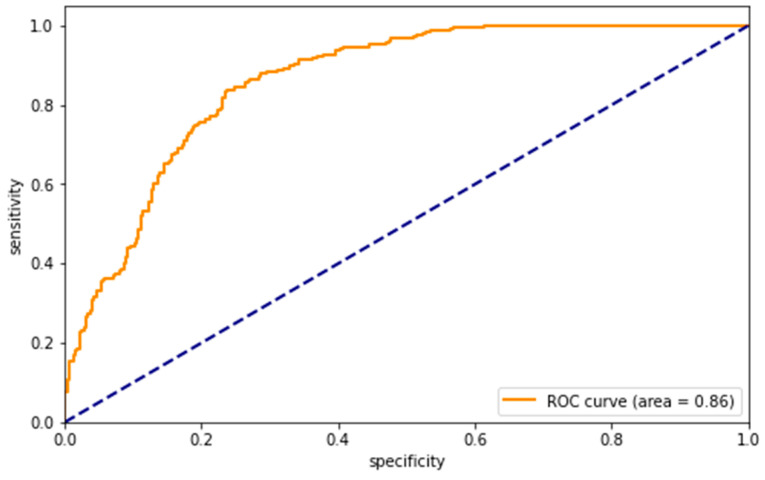
ROC curve of the GBDT model. Note. The blue diagonal line in the figure is the reference line, which represents the result of a classifier using a random guess strategy to classify the samples.

**Figure 2 behavsci-14-00073-f002:**
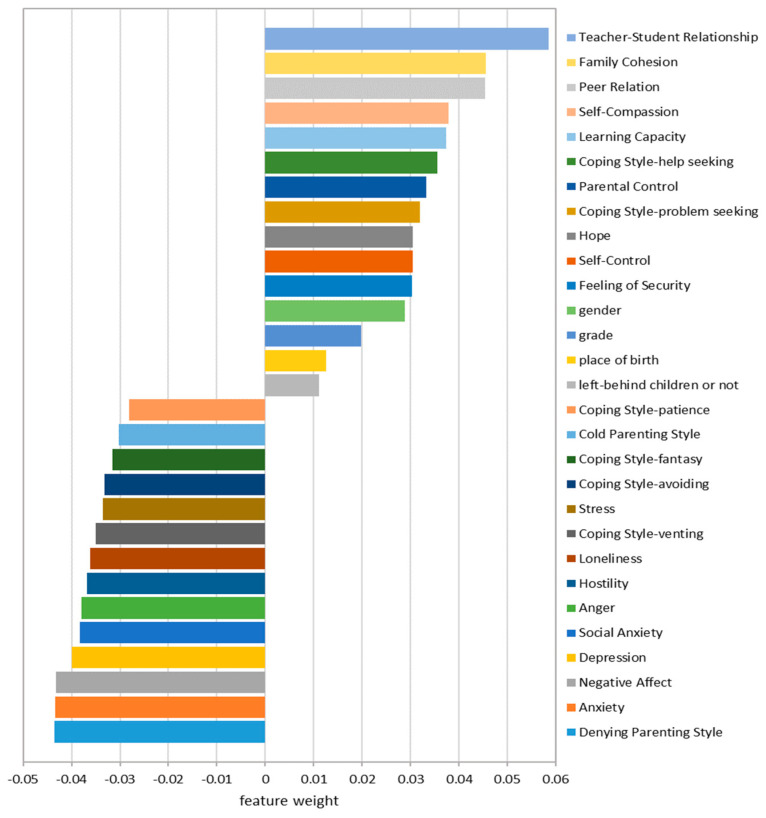
Feature importance for predicting primary and secondary students bullying victimization risk.

**Table 2 behavsci-14-00073-t002:** The individual, family, peer, and school factors between “bullying victimization group” and “not bullying victimization group”.

N = 2767	Bullied (*n* = 1786)	Not Bullied (*n* = 981)	Chi-Square	*t*
Left-Behind Children or Not				
No	37.52%	62.48%	14.019 **	
Yes	29.84%	70.16%		
Grade				
5	41.67%	58.33%	4.755	
6	35.94%	64.06%		
7	36.89%	63.11%		
8	33.79%	66.21%		
9	35.39%	64.61%		
Gender				
Male	30.05%	69.95%	29.296 **	
Female	39.86%	60.14%		
Place of Birth				
Urban Areas	43.74%	56.26%	15.263 **	
Rural Areas	34.05%	65.95%		
Coping Style—problem seeking	3.00 (0.75)	3.13 (0.78)		4.36 **
Coping Style—help seeking	2.68 (0.72)	2.73 (0.72)		1.71
Coping Style—avoiding	2.75 (0.68)	2.74 (0.69)		−0.44
Coping Style—venting	2.41 (0.73)	2.35 (0.72)		−1.97 *
Coping Style—fantasy	2.43 (1.02)	2.27 (1.02)		−3.86 **
Coping Style—patience	3.03 (0.87)	2.93 (0.88)		−3.05 **
Self-Compassion	3.09 (0.47)	3.20 (0.50)		5.61 **
Hope	3.21 (0.91)	3.36 (0.98)		3.96 **
Family Cohesion	3.29 (0.62)	3.45 (0.70)		6.37 **
Parental Control	3.41 (0.84)	3.58 (0.90)		4.95 **
Cold Parenting Style	1.80 (0.66)	1.60 (0.64)		−7.60 **
Denying Parenting Style	1.79 (0.73)	1.55 (0.66)		−8.58 **
Negative Affect	3.18 (0.47)	3.37 (0.47)		9.84 **
Depression	2.01 (0.65)	1.77 (0.61)		−9.48 **
Anxiety	1.83 (0.62)	1.63 (0.58)		−8.53 **
Stress	1.74 (0.67)	1.53 (0.61)		−7.94 **
Feeling of Security	2.65 (0.83)	2.39 (0.83)		−7.93 **
Anger	2.59 (1.00)	2.40 (1.00)		−4.65 **
Hostility	2.50 (0.90)	2.17 (0.86)		−9.65 **
Self-Control	3.17 (0.58)	3.32 (0.59)		6.51 **
Loneliness	2.31 (0.70)	2.08 (0.67)		−8.55 **
Social Anxiety	2.79 (1.19)	2.39 (1.17)		−8.55 **
Peer Relationship	3.35 (0.89)	3.55 (0.93)		5.50 **
Teacher–Student Relationship	2.01 (0.72)	1.77 (0.70)		−8.71 **
Learning Capacity	3.02 (0.67)	3.19 (0.71)		6.17 **

Note. ** *p* < 0.01; * *p* < 0.05.

**Table 3 behavsci-14-00073-t003:** GBDT and logistic regression performances of predicting bullying victimization.

Method	AUC Score	Accuracy	Precision	Recall	F1 Score
GBDT	0.86	0.79	0.78	0.81	0.80
Logistic Regression	0.67	0.67	0.69	0.89	0.78

## Data Availability

The datasets analyzed during the current study are available from the corresponding author on reasonable request.

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
