# Peer review of "Predicting Risk of Bullying Victimization among Primary and Secondary School Students: Based on a Machine Learning Model"

_behavsci, 2024, doi:10.3390/bs14010073_

Round 1
Reviewer 1 Report
Comments and Suggestions for Authors
Dear Authors
Please find my remarks in the document attached.

Author Response
Dear Reviewer:
Please find our responses in the document attached.

Reviewer 2 Report
Comments and Suggestions for Authors
This paper explores potential risk and protective factors that can effectively predict school bullying victimization applying a machine learning analytic approach and examines their importance on school bullying. I have some questions and comments that the authors may consider to revise their manuscript.
First, the strengths of GBDT analysis method should be more thoroughly addressed, especially regarding the aspect that how its application can contribute to the existing school bullying literature. I would suggest that the authors consider conducting additional sensitivity analysis and reporting some differences between the results of the GBDT and those of traditional regression-based models, showing that which one performs better and whether influential predictors are differently selected across the two different analytic models.
Second, according to previous meta-analyses on school bullying, they suggest that prior bullying perpetration is one of the strongest correlates of later bullying victimization, yet this study does not take it into account. I think that an appropriate rationale for this omission seems necessary or this should be discussed as one of the limitations of the study.
Third, more detailed information about sampling and participants is needed. Given that T1 was in November 2020, which was in the middle of COVID-19 pandemic, it's important to know how the survey was conducted, for example, whether it was online or in-person interview and how specifically the survey was conducted such as procedural processes of the survey. Plus, the exact timing of T2 was missing, so this must be reported too. In terms of their sampling method, information about the selection of schools and students was missing, so more details about their sampling strategy should be addressed as well.
Fourth, in the results, the authors claimed that they focus on the top six predictors, but they did not address the specific criteria for selecting them. Relatedly, the authors suggested feature importance scores for each predictor, but what is the meaning of these scores? I understand they are the six highest scores, but still, the authors should discuss the practical meaning of them and magnitudes of their importance, comparing them to those of other predictors.
Fifth, the Discussion section presents feature importance scores of each predictor and their relative magnitudes (in Lines 322 to 352), but it might be more appropriate to address this in the Results section. The authors should focus more on the meaning of the results in the Discussion section.
Comments on the Quality of English LanguageI found too many typos. They should be corrected.
Author Response

(The authors gave the same response as above.)

Round 2
Reviewer 1 Report
Comments and Suggestions for Authors
Dear Authots
Thank you for your effort with the corrections and additions - generally I think the tex is suitable for publication. Good luck!